# A New Restoration Method for Radio Frequency Interference Effects on AMSR-2 over North America

**Wangbin Shen [1], Zhengkun Qin [1,2,*] and Zhaohui Lin [2,3]** 

[1] Center of Data Assimilation for Research and Application, Nanjing University of Information Science & Technology, Nanjing 210044, China; wangbinshen@nuist.edu.cn

[2] Collaborative Innovation Center on Forecast and Evaluation of Meteorological Disasters, Nanjing University of Information Science and Technology, Nanjing 210044, China; lzh@mail.iap.ac.cn

[3] International Center for Climate and Environment Sciences, Institute of Atmospheric Physics, Chinese Academy of Sciences, Beijing 100029, China

\* Correspondence: qzk_0@nuist.edu.cn; Tel.: +86-1879-580-3638

**Abstract:** Observations from spaceborne microwave imagers are important sources of land surface information. However, the low-frequency channels of microwave imagers are easily interfered with by active radio signals with similar frequencies. Radio frequency interference (RFI) signals are widely distributed because of the lack of frequency protection, which seriously hinders the application of microwave imager data in data assimilation and retrieval research. In this paper, a new data restoration method is proposed based on principal component analysis (PCA). Both the ideal and real reconstruction experiments show that the new method can effectively repair abnormal observations interfered by RFI compared with the commonly used Cressman interpolation method because observation information over the whole selected domain is used for restoration in the new method, whereas Cressman interpolation uses only a selection of data around the target observation. The observation errors in the data with RFI can be reduced by one order of magnitude by means of the new method and little artificial information is introduced. One-week restoration validation also proves that the new method has a stable accuracy and broad application prospects.

**Keywords:** radio frequency interference; principal component analysis; data restoration

## 1. Introduction

With the development of fine-resolution forecasting, numerical weather prediction has become increasingly dependent on the assimilation of satellite data [1]. As most microwave imager instrument channels have low frequencies, most of the information contained in the observations of microwave imagers comes from the surface. Due to the low-frequency features, land surface information can be observed by microwave imagers even in certain cloudy conditions [2]. All-time and near all-weather detection make microwave imager data unique for the retrieval of surface meteorological parameters [3]. Currently, spaceborne microwave imagers include the Global Precipitation Measurement (GPM) Microwave Imager (GMI) [4], the full polarization microwave radiometer (WindSat) onboard the experimental Coriolis satellite of the U.S. Department of Defense, the Microwave Radiation Imager (MWRI) onboard the second-generation Polar Orbiting Meteorological Satellite FY-3 [5], and the Advanced Microwave Scanning Radiometer 2 (AMSR-2) onboard Japan's Phase I Global Change Observation Satellite for the Water Cycle (GCOM-W1) [6], which are installed on polar-orbiting meteorological satellites to improve the detection ability of surface remote sensing. Measurements over oceans from microwave imagery can be used to retrieve total atmospheric water vapor content [7] and sea surface temperature (SST) [8]. Over land, measurements at low frequencies are typically

used to retrieve surface parameters, such as soil moisture [9], vegetation water content [10], surface temperature [11], and snow cover [12].

AMSR-2 data are widely used in climate change monitoring and data assimilation research and for both military and civilian communication systems. Due to the lack of attention on channel protection, natural passive thermal radiation signals received by spaceborne microwave radiometers from the Earth-atmosphere system are mixed with signals from active sensors used in communication systems. This phenomenon is called radio frequency interference (RFI) [13,14]. Early studies have shown that RFI widely exists in brightness temperature data of low-frequency bands (e.g., the C and X bands) [15,16], and RFI can significantly increase brightness temperatures at certain frequencies, which is particularly serious for spaceborne passive remote sensing instruments with weak signals. The strong signals emitted from these interfering sources conceal relatively weak thermal radiation signals from the Earth-atmosphere system, resulting in the distortion of brightness temperature data from satellite observations, which seriously affects the accuracies of various retrieval products and other applications of these data. Therefore, it is important to correctly identify and remove these interference signals hiding in satellite observations so as to enhance the application value of these signals. Recently, scholars have emphasized and made outstanding contributions to the identification of land RFI. For example, in 2004, Li et al. [13] discovered the phenomenon of RFI in C-band (6.925 GHz) observations from AMSR-E and proposed the spectral difference method to identify regions with moderate or high RFI intensity. In 2006, Li et al. [14] improved the RFI identification algorithm using principal component analysis (PCA), which fully considers observation data correlations between channels. Wu et al. [17] proposed the linear fitting method as a RFI correction algorithm for AMSR-E observation data. To realize the global applicability of RFI detection methods, Zou et al. [18] proposed the normalized PCA method to effectively identify the RFI signal in observations over snow- and ice-covered surfaces. To realize RFI detection in Greenland and South and Arctic regions, Zhao et al. [19] used a dual principal component analysis (DPCA) identification method and successfully identified the RFI signal in WindSat full polarization radiation brightness temperature data at the edge of ice sheets. In addition, Zou et al. [20,21] further proposed an empirical model of interference signal recognition and interference data restoration based on the scintillation angle to solve the problem of satellite TV signal interference in offshore areas. Additionally, many techniques have been developed to identify contaminated data according to the statistical properties of the brightness temperature and their polarimetric properties in SMOS data [22–25]. Although current research has achieved a global identification of RFI signals, studies on RFI detection remain at the qualitative identification stage. Due to the widespread existence of RFI, the application of microwave imager data is greatly restricted, especially for long-term retrieval datasets. RFI causes a wide range of missing data therefore, effectively repairing interfered data has become particularly crucial.

In this paper, after the detection and analysis of the RFI signal in AMSR-2 brightness temperature observations, a new data restoration method is established using the PCA decomposition method. Idealized and realistic data experiments on restoring RFI interference data over the continental U.S. are performed to prove the effectiveness and accuracy of the new method. Finally, a one-week restoration test is conducted to verify the stability of the new method. The structure of the article is as follows: Section 2 briefly describes the data characteristics of AMSR-2, Section 3 introduces the RFI signal detection and restoration method, Section 4 shows the results of repairing the RFI interference data, and Section 5 presents a discussion of the paper. Section 6 concludes this paper.

## 2. AMSR-2 Data

AMSR-2 data were selected to perform RFI detection and restoration tests. AMSR-2 is onboard GCOM-W1, which was developed by the Japan Aerospace Exploration Agency (JAXA) and launched successfully on 18 May 2012. AMSR-2 is a passive microwave remote sensing instrument that cannot emit electromagnetic waves but can detect the characteristics of a target by passively receiving the microwave energy emitted by the observed object. AMSR-2 is a seven-frequency passive microwave

radiometer system that measures brightness temperatures at 6.9, 7.3, 10.7, 18.7, 23.8, 36.5, and 89.0 GHz in horizontal and vertical polarization modes, resulting in a total of 14 observation channels. Each scanning band contains data from the relevant scanning area, which are stored in HDF5, a hierarchical data format with L1R representing the resampled data. The spatial resolution of all channels used in this paper was resampled to $35 \times 62$ km$^2$.

L1R brightness temperature data from AMSR-2 over North America from 1 September to 7 September 2016 were considered in this paper. Figure 1 shows the spatial distribution of the brightness temperature of the AMSR-2 6.9 GHz and 10.7 GHz horizontal polarization channels and their differences at 08:16 UTC on 1 September 2016. The brightness temperatures of the two channels are in the range of 250 to 290 K. The brightness temperature of the 6.9 GHz channel (Figure 1a) is substantially lower than that of the 10.7 GHz channel (Figure 1b) because of its lower frequency. Negative differences should dominate the continent, but many abnormally large positive values are observed (greater than 5 K), as shown in Figure 1c. These large positive differences are located mainly over large cities, such as in California, Washington, Michigan, Ohio, Virginia, and Maryland. According to previous studies of the RFI [13], these large positive differences imply the existence of RFI because RFI significantly increases the brightness temperature observations of low-frequency channels.

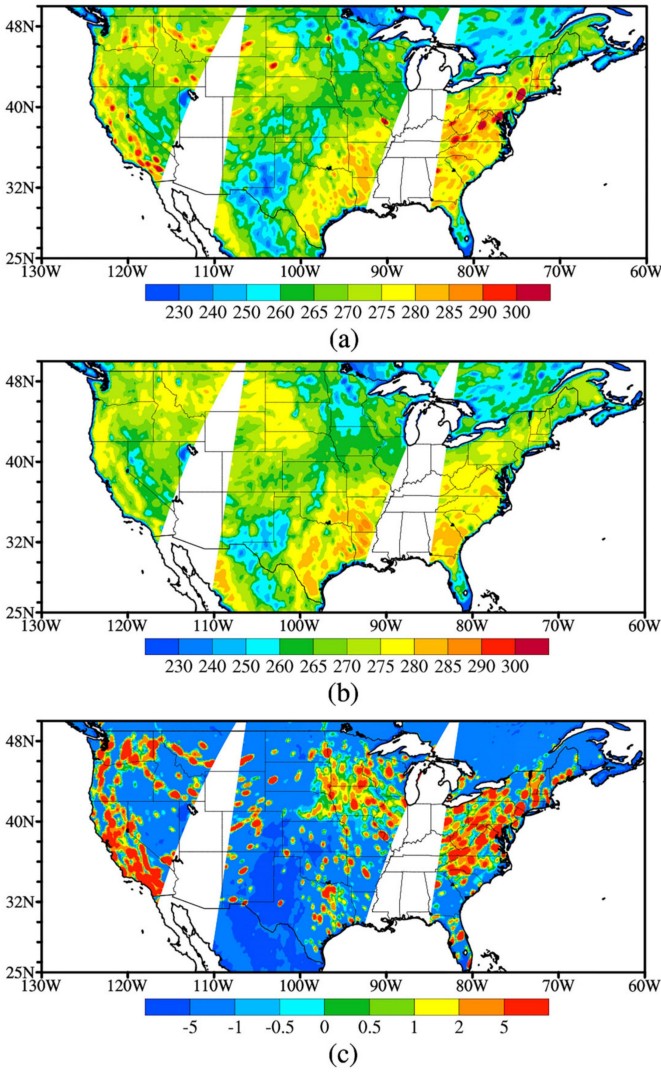

**Figure 1.** (**a**) Spatial distribution of brightness temperature of the 6.9 GHz, (**b**) 10.7 GHz horizontal polarization channels, and (**c**) their differences (6.9–10.7) over the North American continent on 1 September 2016.

## 3. RFI Detection and Restoration Methods

### 3.1. RFI Detection Method

The accurate detection of RFI signals is a prerequisite for subsequent restoration work. Many studies have focused on RFI detection over land [13–15]. Among them are two common methods: The spectral difference method and the spatial decomposition method based on PCA. As the spectral difference detection method is easily disturbed by surface types and sudden changes in temperature, the spatial decomposition method proposed by Zou et al. [18] is used here. First, the interference coefficient matrix is constructed using the normalized brightness temperature difference between a low-frequency channel and a high-frequency channel. By means of the PCA method, the RFI signal can effectively be extracted from the coefficient matrix to the first PCA mode. As the surface characteristics impact the observations of all channels, the normalized interference coefficient matrix can filter the disturbance of special underlying surfaces, such as snow and ice.

Figure 2 shows the distribution of RFI-contaminated data identified by the normalized PCA method for the 6.9 GHz horizontal polarization channel over the United States. As seen from this figure, the first principal component coefficient is typically between −2 and 1.0, and the red area in the figure (i.e., where the coefficient of the first principal component is greater than 2.0) indicates a higher probability of RFI occurrence. The test results (Figure 2) show that most large cities in the United States have pronounced radio interference signals, especially cities along the west coast.

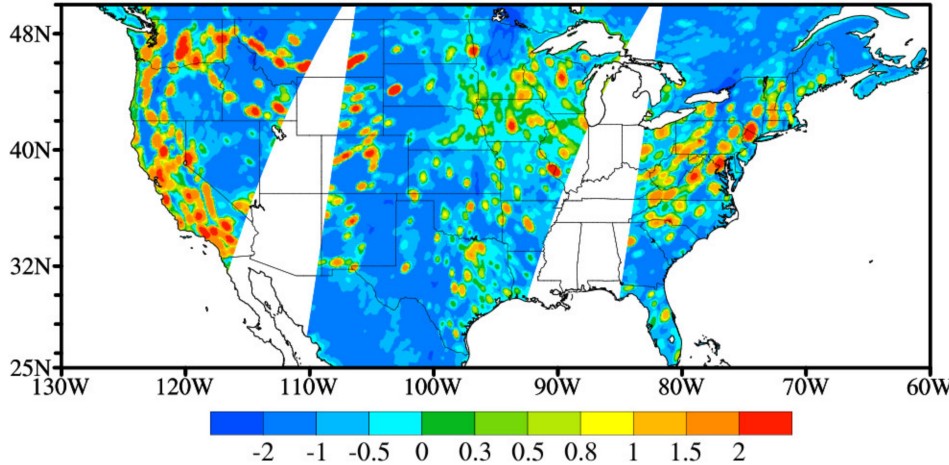

**Figure 2.** Spatial distribution of radio frequency interference signals identified by the normalized PCA method in observations of the 6.9 GHz horizontally polarized channel on 1 September 2016.

The identification results show many interference signals in the AMSR-2 data over North America. These interfered data should be removed before data assimilation and retrieval studies are conducted, which results in a considerable waste of data and a large amount of missing data in retrieved datasets. Therefore, after the effective detection of RFI signals, it is essential to restore these abnormalities. Based on the modal orthogonality of PCA, a step-by-step iterative restoration method is proposed to restore these interfered observations.

### 3.2. RFI Restoration Method: Principle of the PCA Iterative Method

Many existing interpolation algorithms, such as the Cressman interpolation method, can be used to reconstruct the missing measurements. However, methods such as the Cressman interpolation method use information from only a few observations around the target. Thus, it is easy to introduce artificial errors due to some small-scale extremes, which are particularly numerous for surface air temperatures. When a large number of observations are missing, interpolation results often fail to accurately represent the spatial continuity of data.

In contrast to the Cressman interpolation method, PCA has the advantage of using term space reduction technology to decompose the spatiotemporal structure of the original variable field in an orthogonal manner, thus obtaining a few uncorrelated typical modes to represent the main features of the original variable field. There are two important characteristics of these modes extracted by PCA. First, a mode is determined after considering the spatiotemporal variation characteristics of variable fields and the mode is not affected by a few extreme values. The second characteristic is that each mode extracted by PCA is orthogonal to the others therefore, modes and their corresponding coefficients can be recovered independently, and the coefficients of each mode can be determined via an iterative step-by-step process without considering the interaction between modes. Based on these two characteristics, we attempted to establish a PCA-based restoration method to recover data with RFI, and the implementation of the method is described as follows.

Assuming that pixels contaminated by RFI have been effectively detected, without loss of generality, we attempted to repair contaminated pixel *p*. It is noted that if the pixels interfered by RFI were not accurately detected, these outliers with abnormal high values around the reconstruction pixel would make the reconstruction of the original values higher than the real value. Thus, the accurate detection of RFI signals plays a key role in the reconstruction.

First, the data matrix for PCA analysis was established. Assuming that pixel *p* was located in the *j*th FOV of the *i*th scan line, the 600 pixels closest to pixel p in space were selected. Then, the RFI-affected data detected by normalized PCA were removed, and the number of remaining observation points were denoted as *n*. For the n selected observation points, the data of 9 channels for each observation point were selected. Assuming that the data of the horizontal polarization channel of 6.9 GHz would be repaired, the selected 9 channels included the 6.9 GHz horizontal polarization channel, as well as the 10.7, 18.7, 23.8, and 36.5 GHz horizontal and vertical polarization channels, denoted as $TB_{6H}$, $TB_{10H}$, $TB_{10V}$, $TB_{18H}$, $TB_{18V}$, $TB_{23H}$, $TB_{23V}$, $TB_{36H}$, and $TB_{36V}$. Here, TB stands for brightness temperature, and H and V denote horizontal and vertical polarization, respectively.

The final data matrix used for repair can be expressed as $A_{m \times n}$:

$$
A_{m \times n} = 
\begin{bmatrix}
TB_{6H,1} & TB_{6H,2} & \cdots & TB_{6H,p} & \cdots & TB_{6H,n} \\
TB_{10H,1} & TB_{10H,2} & \cdots & TB_{10H,p} & \cdots & TB_{10H,n} \\
\vdots & \vdots & \vdots & \ddots & \vdots & \vdots \\
TB_{36H,1} & TB_{36H,2} & \cdots & TB_{36H,p} & \cdots & TB_{36H,n} \\
TB_{36V,1} & TB_{36V,2} & \cdots & TB_{36V,p} & \cdots & TB_{36V,n}
\end{bmatrix}
$$

$$
=
\begin{bmatrix}
TB_{1,1} & TB_{1,2} & \cdots & TB_{1,p} & \cdots & TB_{1,n} \\
TB_{2,1} & TB_{2,2} & \cdots & TB_{2,p} & \cdots & TB_{2,n} \\
\vdots & \vdots & \vdots & \ddots & \vdots & \vdots \\
TB_{8,1} & TB_{8,2} & \cdots & TB_{8,p} & \cdots & TB_{8,n} \\
TB_{9,1} & TB_{9,2} & \cdots & TB_{9,p} & \cdots & TB_{m,n}
\end{bmatrix},
\tag{1}
$$

where *m* is 9, which represents the selected 9 channels. $TB_{6H,p}$ is the brightness temperature of the 6.9 GHz horizontal polarization channel for pixel *p* affected by the RFI. Initially, we set $TB_{6H,p} = 0$. By applying PCA decomposition to matrix A, we obtained:

$$
A_{m \times n} = V_{m \times m} Z_{m \times n} = \sum_{k=1}^{m} \vec{V}_k^T \vec{Z}_k = \sum_{i=1}^{m} \sum_{j=1}^{n} \sum_{k=1}^{m} v_{i,k} z_{k,j},
\tag{2}
$$

where $\vec{V}_k$ is generally referred to as the mode vector and $\vec{Z}_k$ is the coefficient vector. *k* represents the *k*th mode. Since the number of pixels selected is greater than the number of channels, the maximum number of modes obtained by PCA is *m*. Similarly, PCA is computed for each contaminated pixel using the surrounding pixels without the interference of RFI.

Since the first mode of PCA contains the average spatial features of the data [26], it is reasonable to believe that the first mode does not include the interference information of the abnormal pixel, so the first mode can be used to reconstruct:

$$TB^1_{1,p} = v_{1,1}z_{1,p}, \tag{3}$$

Obtaining accurate first-mode information is a prerequisite to ensure the repair effect of the method. Although information of a value of zero is not included in the first mode, some useful information belonging to the first mode may disperse to other modes. Therefore, it is necessary to gradually gather other average characteristics via the iteration method.

After obtaining the value of $TB^1_{1,P}$, this value was substituted into the original data matrix to form a new data matrix $A^1$:

$$A^1 = \begin{bmatrix} TB_{1,1} & TB_{1,2} & \dots & TB^1_{1,P} & \dots & TB_{1,n} \\ TB_{2,1} & TB_{2,2} & \dots & TB_{2,p} & \dots & TB_{2,n} \\ \vdots & \vdots & \vdots & \ddots & \vdots & \vdots \\ TB_{8,1} & TB_{8,2} & \dots & TB_{8,p} & \dots & TB_{8,n} \\ TB_{9,1} & TB_{9,2} & \dots & TB_{9,p} & \dots & TB_{9,n} \end{bmatrix}, \tag{4}$$

and applying PCA decomposition to matrix $A^1$ again yielded:

$$A^1 = V^1Z^1 = \sum_{i=1}^{m} \sum_{j=1}^{n} \sum_{k=1}^{m} v^1_{i,k}z^1_{k,j}. \tag{5}$$

The new recovered brightness temperature of pixel *p* based on the first mode was obtained as follows:

$$TB^2_{1,p} = v^1_{1,1}z^1_{1,p}. \tag{6}$$

In matrix $A^1$, the reconstructed data of mode one were used to replace the zero value in matrix A to reduce the influence of abnormal data on the integrity of mode one. Therefore, the PCA mode one of $A^1$ was closer to the real mode one, that is, the first PCA mode obtained without the influence of RFI. Due to the uniqueness of the truth value, the first mode of PCA should gradually stabilize over iterations. Therefore, the variation in reconstruction values obtained by iteration can be used as an indicator to stop the iteration process. We calculated the absolute value of the difference between the previous and current reconstruction of the p-point 6.9 GHz horizontal channel brightness temperature, $\left| TB^1_{1,p} - TB^2_{1,p} \right|$. If the difference was greater than a specified threshold, useful information remained in other modes, and the above process was repeated. A new data matrix A was constructed and PCA decomposition was conducted to establish the brightness temperature of the first-mode reconstruction until the difference was less than the specified threshold. The requirements were assumed to be satisfied after $t_1$ iterations, that is:

$$\left| TB^1_{1,p} - TB^2_{1,p} \right| \leq \varepsilon. \tag{7}$$

The average characteristic represented by the first mode is completely preserved, which means the average feature of the brightness temperature of the 6.9 GHz horizontal channel of the pixel was also well reproduced. In this paper, we empirically specified $\varepsilon$ as 0.01. The five-pointed star in Figure 3a represent the point needs the reconstruction, the color dots represent observations used in the reconstruction, and it can be seen that many of the surrounding pixels were removed because of identified RFIs. Figure 3b shows the changing curve for $\left| TB^{t_1-1}_{1,p} - TB^{t_1}_{1,p} \right|$ of each mode during iterative reconstruction varying with the number of iterations. After 5 iterations, the value of $\left| TB^{t_1-1}_{1,p} - TB^{t_1}_{1,p} \right|$ gradually approached zero.

Varying with the number of iterations for the iterative reconstruction of the first PCA mode.

Based on the complete reproduction of the first-mode features of the 6.9 GHz horizontal channel on the *p*-point, we continued to extract information from the second PCA mode. After $t_1$ iterations, the new data matrix is $A^{t_1}$:

$$A^{t_1} = \begin{bmatrix} TB_{1,1} & TB_{1,2} & \ldots & TB_{1,P}^{t_1} & \ldots & TB_{1,n} \\ TB_{2,1} & TB_{2,2} & \ldots & TB_{2,p} & \ldots & TB_{2,n} \\ \vdots & \vdots & \vdots & \ddots & \vdots & \vdots \\ TB_{8,1} & TB_{8,2} & \ldots & TB_{8,p} & \ldots & TB_{8,n} \\ TB_{9,1} & TB_{9,2} & \ldots & TB_{9,p} & \ldots & TB_{9,n} \end{bmatrix}, \tag{8}$$

PCA decomposition is applied to matrix $A^{t1}$, and the brightness temperature of the 6.9 GHz-H channel was reconstructed on the *p*-point using the first two modes:

$$TB_{1,p}^{t_1+1} = \sum_{k=1}^{2} v_{1,k}^{t_1} z_{k,p}^{t_1}. \tag{9}$$

From the previous analysis, we obtained stable first-mode information and constructed a new data matrix, which is equivalent to setting the other mode components, except the first mode, to zero. The above iterative process can be repeated to obtain accurate information with the first two modes. Finally, the restoration ends when all PCA modes and their corresponding coefficients are included. Similarly, we can obtain as much accurate observation information as possible with the constraints of peripheral observations of the 6.9 GHz-H channel and undisturbed observations of other channels.

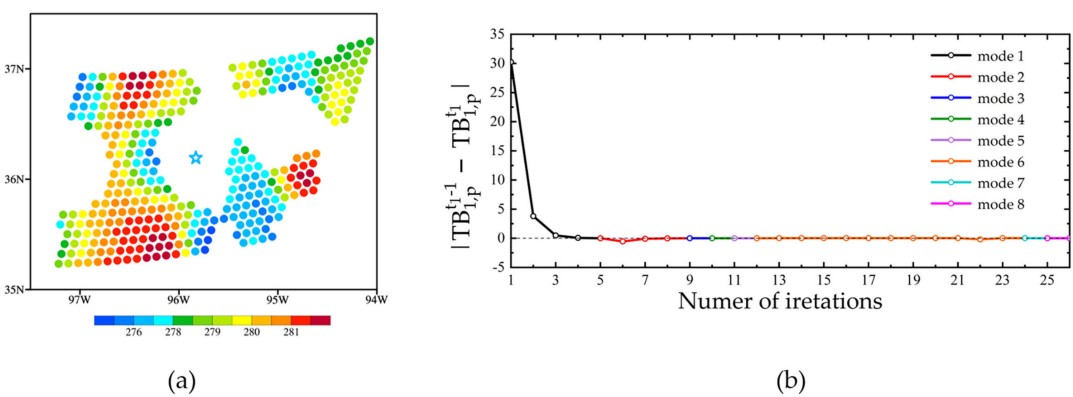

(a)                                                                 (b)

**Figure 3.** (**a**) Brightness temperature distribution after removing the affected point, (**b**) the difference between the previous and current reconstruction of the p-point varying with the number of iterations of the first PCA mode.

## 4. Results

### 4.1. Ideal Test Results

The true values of data contaminated by RFI are impossible to obtain. Thus, an ideal test was conducted to evaluate the effectiveness and accuracy of the proposed method based on data from the 36 GHz horizontal polarization channel that are free from RFI. To ensure that the ideal test was realistic, the pixels corresponding to locations of RFI of the 6 GHz-H channel were removed before restoration. The five-pointed star in Figure 4a–c represent the point used for reconstruction. After the iterative process, the BT of reconstruction was very close to the observation, which indicates that the PCA iterative repair method is feasible and can restore the original data with high precision, even if there was a large number of missing values around the target observation.

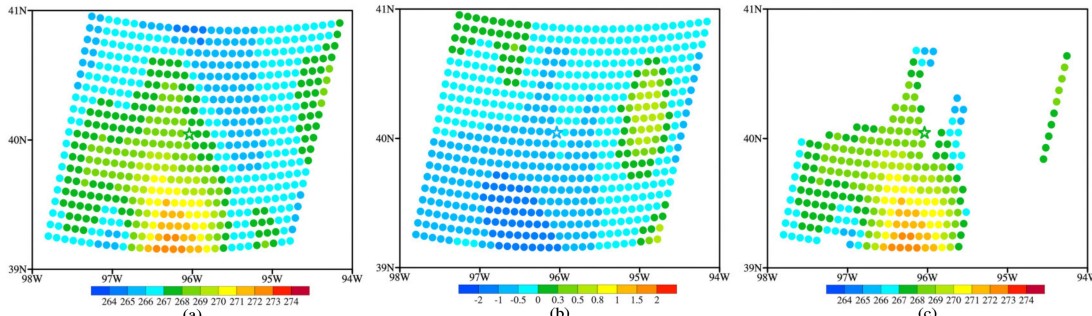

**Figure 4.** (**a**) Distribution of the observed brightness temperature of the 36 GHz-H channel, (**b**) radio frequency interference (RFI) signal distribution of the 6 GHz-H channel, and (**c**) brightness temperature distribution after removing the affected point.

Figure 5 shows the differences in the 36 GHz-H brightness temperature between the original observations and those of the restorations after iterations of eight modes. After the first five iterations, the first mode converged. Moreover, the iterative values of the latter modes gradually converged toward the true value and the final reconstruction error was less than 0.2 K, indicating that the iterative method is feasible, with fast iterative convergence and high reconstruction efficiency.

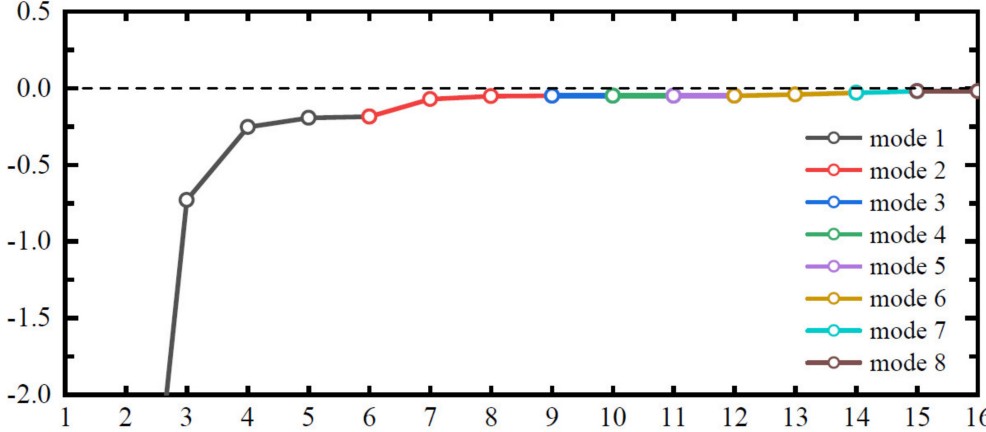

**Figure 5.** The brightness temperature difference between restoration values with gradually increased modes included and the original observation.

Result for an individual pixel is far from enough to characterize the error of the new method, so the same iteration test was performed for each observation of 36 GHz-H. Thus, there were sufficient samples to further verify the feasibility of the PCA iterative method and to analyze the error characteristics of the method. Identical to the test of individual pixels, pixels with 6 GHz-H data identified as contaminated data were removed before restoration to ensure the feasibility of the method to the real condition. Figure 6a shows the distribution of the 36 GHz-H brightness temperature observations, Figure 6b shows the reconstruction results of the PCA iterative method, and Figure 6c shows the difference between the observations and reconstructed data (observation-reconstructed). For comparison, a linear form with dual polarizations was used to predict the 36 GHz-H brightness temperature from 23 GHz channels as described by Wu et al. [17]. The linear fitting form is as follows:

$$TB_{H,36} = 64.49462 + 1.59037 \times TB_{H,23} - 0.82308 \times TB_{V,23} \tag{10}$$

The reconstruction and difference results are showed in Figure 6d–e. We also present the reconstruction and difference results of the Cressman interpolation method in Figure 6f–g. Comparison of the three methods shows that the mean difference between the observed brightness temperature

and the PCA iterative reconstruction temperature was less than 0.5 K, and the root mean square error was 0.36, while the root mean square error of linear fitting method was 2.86K which is greater than that of the PCA iterative method, whereas the errors of the Cressman interpolation method were much larger than those of the PCA iterative method. Many of the errors were larger than 5 K, particularly in areas with missing observations, such as the west coast of the United States, where the ratio of missing data was close to 30%. The traditional interpolation method had difficulty achieving an ideal restoration effect because of the weak continuity of observations over these areas. The linear fitting method was based on the principle of high correlations between different channels [17], when the RFI was detected for one channel, the brightness temperature of another channel with the closest frequency was used to simulate the contaminated one by the linear regression method. The linear regression method only used observations from one single pixel and restored the contaminated data based only on observation information from other channels. By contrast, the PCA iterative algorithm extracts the physical characteristics of the data using the whole domain data and reconstructs the missing physical observation data by iteration, which guarantees that the physical continuity of observations remained unchanged.

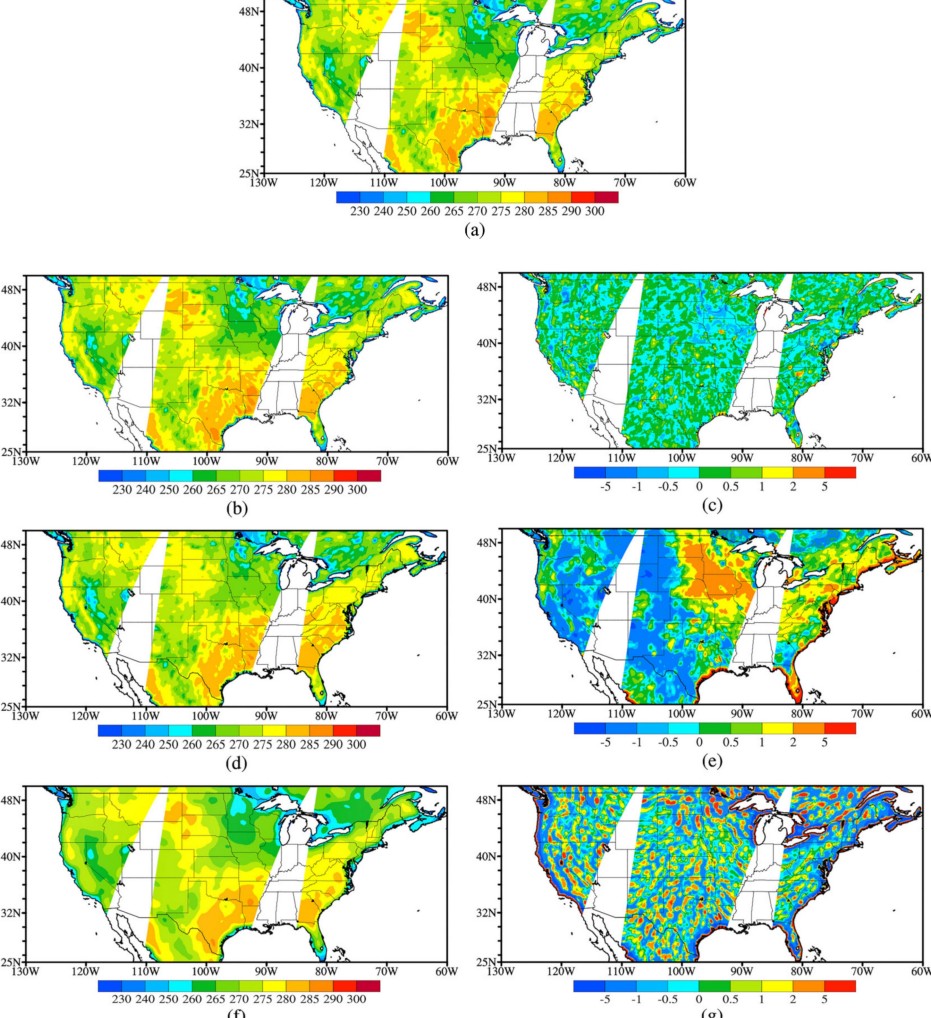

**Figure 6.** (**a**) Distribution of the observed brightness temperature of the 36 GHz-H channel, (**b**) reconstructed brightness temperature of the 36 GHz-H via the PCA iterative method, (**c**) for (**b**)–(**a**), (**d**) reconstructed brightness temperature of the 36 GHz-H via the linear fitting method, (**e**) for (**d**)–(**a**), (**f**) reconstructed brightness temperature of the 36 GHz-H by the Cressman interpolation method, and (**g**) for (**f**)–(**a**).

## 4.2. Analysis of the Error Characteristics of the Reconstruction Methods

Figure 7 is a scatter plot of the restorations of the PCA iterative method and those of the linear fitting method as well as the Cressman interpolation method. In the figure, the x-axis represents the observed brightness temperature of the 36 GHz-H channel and the y-axis represents the restored brightness temperature of the three reconstruction methods. The PCA iterative reconstruction temperature is very close to the observations, while the data reconstructed by the linear fitting method were away from the observations. But the data reconstructed by the Cressman method were far from the observations, which indicates that the reconstruction errors of the new method were smaller than those of the linear fitting method and Cressman interpolation method.

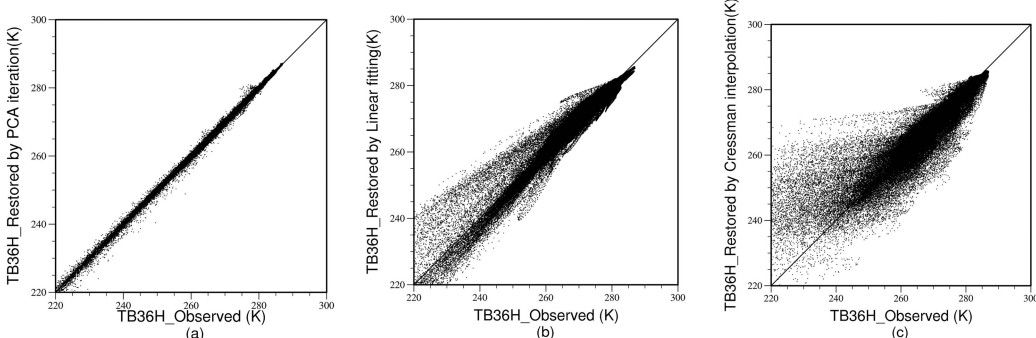

**Figure 7.** The brightness temperature scatter plots of the 36 GHz-H channel between observations (x-axis) and data reconstructed by (**a**) the PCA iterative method (y-axis), (**b**) the linear fitting method (y-axis), and (**c**) the Cressman interpolation method (y-axis).

To verify that the iterative method did not produce over-repair effects, the probability density distribution (PDF) of the differences between the observations and the data reconstructed by the three methods was calculated. Figure 8 shows the PDF of the difference between the observed brightness temperature and the restored brightness temperature. The skewness of the PCA iterative method was 0.035, which is very close to that of the unbiased distribution, so the method did not produce excessive repair. Furthermore, the kurtosis coefficient was 3.984, which was higher than that of the linear fitting method and the Cressman interpolation method. This result also demonstrates that the accuracy of the new method was higher than that of the Cressman method. The skewness of the linear fitting method was 3.03 while it was 3.73 of the Cressman interpolation method. The large positive skewness suggests that the brightness temperatures reconstructed by the two methods were more likely to be affected by high brightness temperatures.

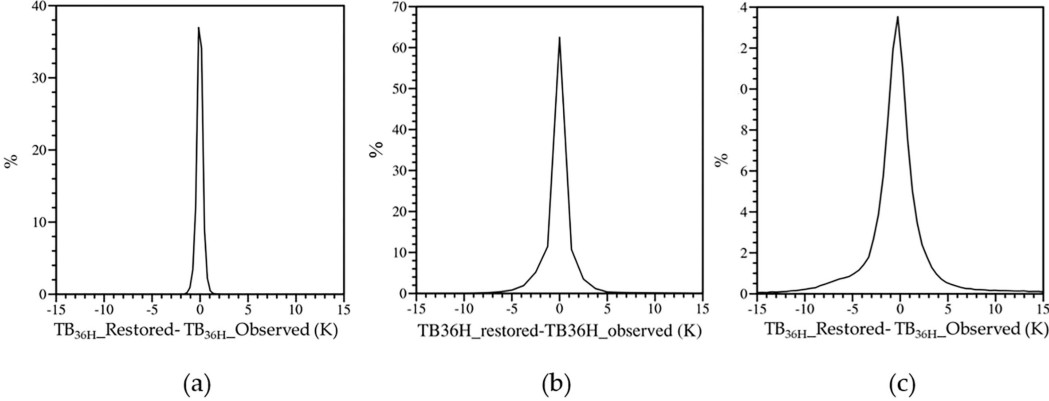

**Figure 8.** The probability density distribution (PDFs) of the difference between observed brightness temperature and restored brightness temperature for the PCA iterative method (**a**), linear fitting method (**b**) and the Cressman interpolation method (**c**).

### 4.3. Analysis of the 6 GHz-H Channel Repair Results

The PCA iterative restoration method has higher accuracy and more effectiveness than the linear fitting method and Cressman method in ideal experiments. Next, we conducted a real repair experiment on the 6.9 GHz horizontally polarized channel.

The normalized principal component analysis (NPCA) method was used to indicate the positions of observed brightness temperatures interfered by RFI. Those RFI points were then reconstructed by the PCA iterative method and the linear fitting method as well as the Cressman interpolation method. The brightness temperature distribution after repair is presented in Figure 9, which shows that the abnormally high values were restored by the PCA iterative reconstruction and the brightness temperature distribution obtained by reconstruction conformed to the characteristics of natural surface emission.

After correcting the abnormal values, it is necessary to verify that the corrected brightness temperature is reasonable. Although there were no real observations for comparison, the high spatial correlation of brightness temperatures between different channels was a good validation metric. Therefore, the brightness temperatures of the 10 GHz-H channel were selected to verify whether the channel correlation was retained by the new method. Figure 10 shows a scatter plot of the brightness temperatures of the 10 GHz-H channel and those of the 6 GHz-H channel. There were many abnormally high values of observed brightness temperature in the 6 GHz-H channel, even exceeding the dynamic upper limit of the instrument of 350 K. However, these abnormally high values disappeared after restoration. Moreover, the brightness temperatures restored by the PCA iterative method maintained a high correlation with the 10 GHz-H brightness temperatures, whereas the brightness temperatures obtained by the Cressman interpolation method could not maintain a good consistency with the brightness temperatures of the 10 GHz-H channel. However, there were still some high values in the brightness temperatures reconstructed by the linear fitting method and the Cressman interpolation method.

The NPCA method was used to verify whether RFI was eliminated from the repaired results. Figure 11 shows the distribution of RFI signals identified by NPCA for data recovered by the PCA iteration method and the linear method as well as the Cressman method for the 6 GHz-H channel. As mentioned above, high values indicate the presence of RFI at 6.925 GHz. The RFI value identified by the NPCA method for brightness temperatures restored by the PCA iteration method was generally lower than that for the Cressman method and it can been seen that some high values still remained in the results of linear fitting method. Therefore, the brightness temperature restored by the PCA iteration method is more reliable.

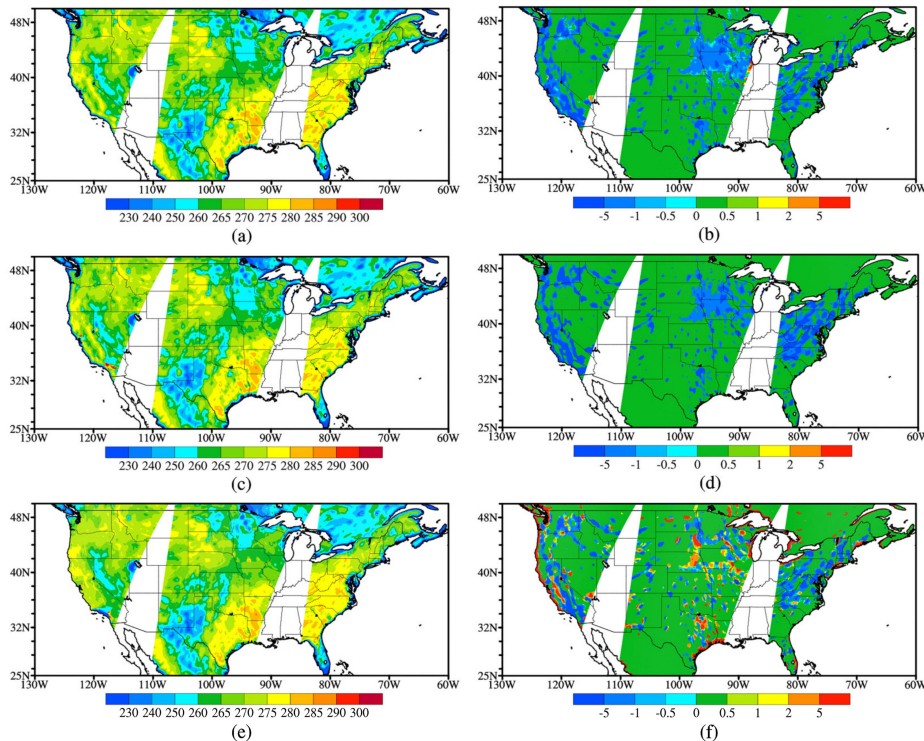

**Figure 9.** (**a**) The brightness temperature of 6 GHz-H restored by the PCA iterative method and (**b**) its difference from the observed brightness temperature. (**c**) The 6 GHz-H brightness temperature restored by the linear fitting method and (**d**) its difference from the observed brightness temperature. (**e**) The brightness temperature of 6 GHz-H restored by the Cressman interpolation method and (**f**) its difference from the observed brightness temperature.

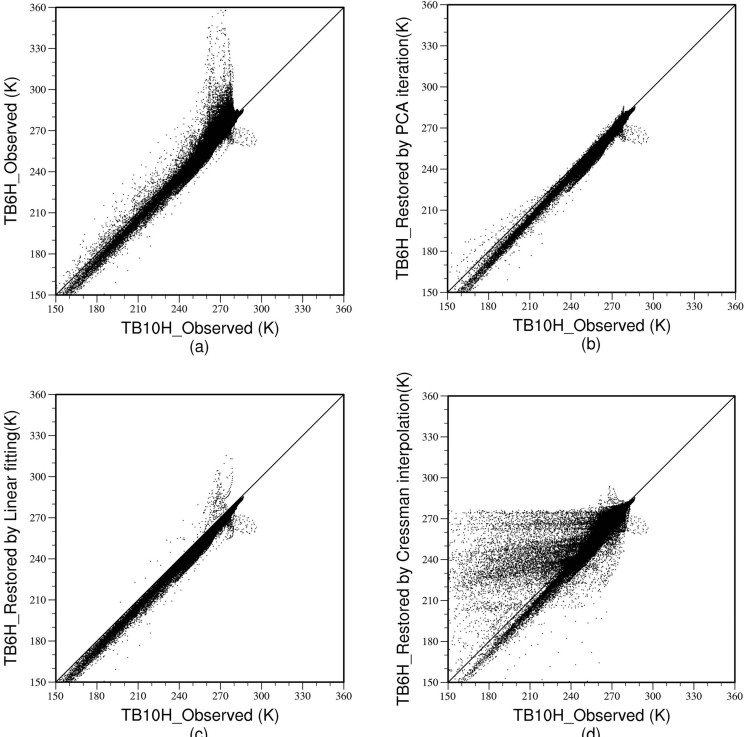

**Figure 10.** Scatter plots of observed brightness temperatures of the 10 GHz-H and 6 GHz-H (**a**) channels and brightness temperatures restored by: (**b**) The PCA iteration method, (**c**) the linear fitting method, and (**d**) and the Cressman interpolation method.

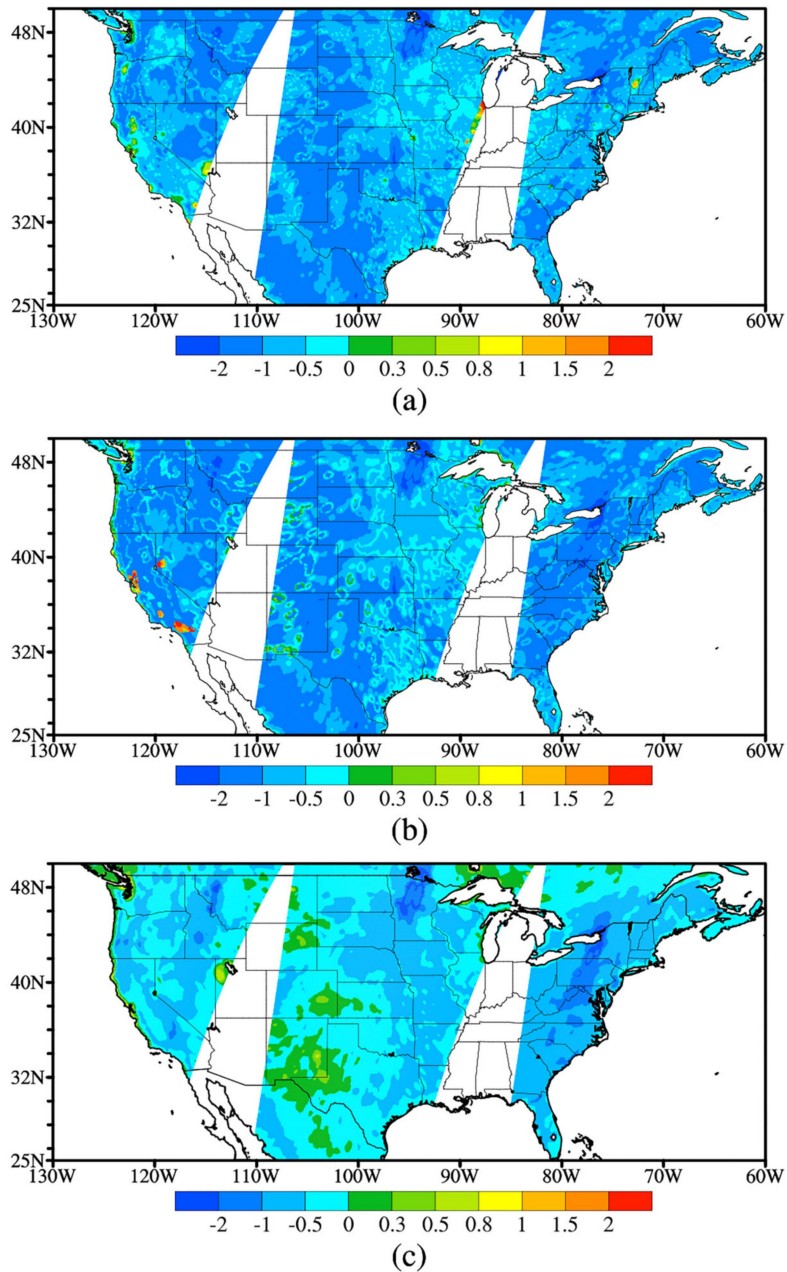

**Figure 11.** Distribution of RFI signals identified by normalized principal component analysis (NPCA) for data recovered by (**a**) the PCA iteration method, (**b**) the linear fitting method for the 6 GHz-H channel, and (**c**) the Cressman method for the 6 GHz-H channel.

### 4.4. Restoration Results of Multitime Observation

One-week AMSR-2 observations were selected to verify and prove the stability of the new restoration method.

The following figure shows the restoration results of all polarization of 6 GHz on September 5 2016 and September 7 2016. As shown in Figure 12, many areas with abnormally high brightness temperature values that far exceeded 300 K exist in the 6 GHz-H channel. The abnormally high values were eliminated and the brightness temperature distribution was reasonable after the correction.

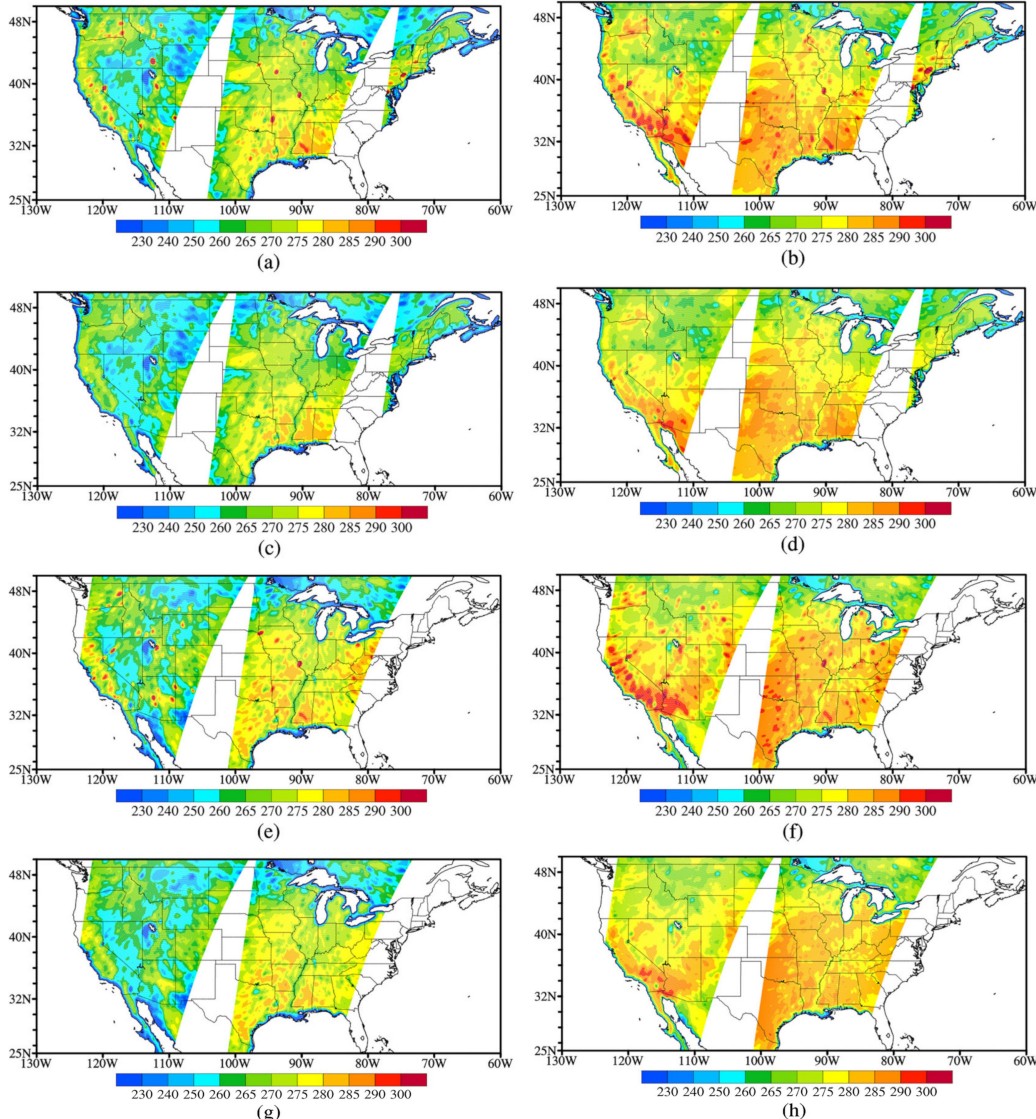

**Figure 12.** The spatial distribution of the observed (**a**,**b**,**e**,**f**) and restored (**c**,**d**,**g**,**h**) brightness temperatures at 6 GHz-H (**a**,**c**,**e**,**g**) and 6 GHz-V (**b**,**d**,**f**,**h**) on 5 September 2016 (**a**–**d**) and 7 September 2016 (**e**–**h**).

Figure 13 shows the PDF of the difference between the reconstructed brightness temperatures and the original observations of all polarization of 6 GHz from September 1 to September 7. The skewness of the PDF for 6 GHz-H was −5.7, while it was −6.87 for 6 GHz-V, which means that the new brightness temperature was generally lower than the observed value. In other words, the abnormally high brightness temperatures caused by RFI were successfully restored.

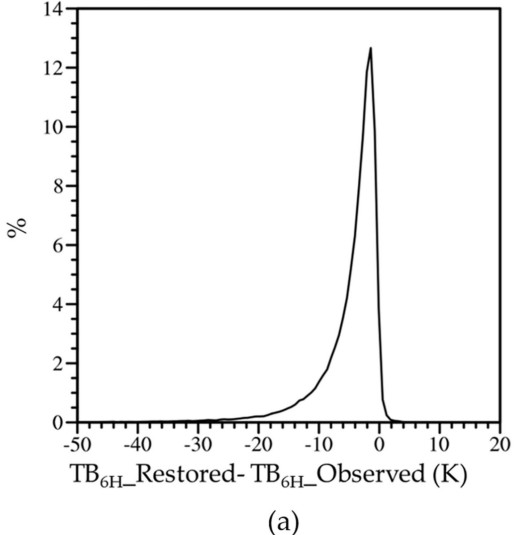
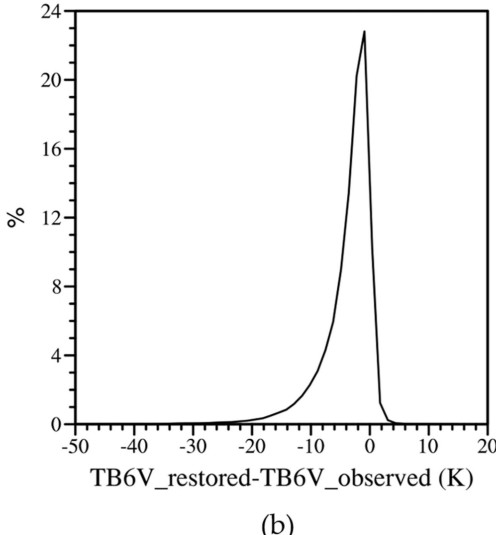

(a)                                                 (b)

**Figure 13.** The PDF of the difference between observations with RFI and the restored brightness temperatures of 6 GHz-H (**a**) and 6 GHz-V (**b**) during 1–7 September 2016.

## 5. Discussion

Currently, the contaminated data could not be used effectively due to the RFI. However, future sensors need to use these unprotected bands in order to achieve a specific observation target. This results in fueling an imperative research into RFI restoration in spaceborne microwave radiometer measurements. Repairing the RFI contaminated data is of great significance to evaluating the accuracy of the inversion of atmospheric and surface parameters of microwave data. The results prove that the new method has good stability and prospects for long-term RFI data restoration. Therefore, more research about how to use the recovered data in data assimilation and retrieval research will be conducted and the accuracy of the recovered data also require further research.

## 6. Conclusions

Due to the lack of frequency protection, the low-frequency channels of AMSR-2 were often disturbed by active RFI, especially observations over large cities. Many studies have focused on RFI detection, but because it is impossible to remove RFI signals in most cases, contaminated data must be removed before AMSR-2 data can be used for data assimilation and retrieval applications. This process results in a large quantity of missing data, which is particularly detrimental for climate applications of AMSR-2 observations.

RFI-contaminated data are difficult to recover because low-frequency channels are window channels, and their observations are significantly influenced by various kinds of surface information. The variation in land type causes dramatic surface emissivity changes. Moreover, surface weather is dominated by small-scale weather systems, which result in substantial errors for regular interpolation methods, such as the Cressman interpolation method, because most of these methods make use of only a few observations close to the target position. Wide range missing data and the extreme values of small-scale weather systems cause considerable difficulties for regular interpolation methods. A new restoration method based on iterative PCA analysis was proposed in this paper to recover RFI-contaminated observations over land. Based on the effective detection of RFI signals using a normalized PCA method developed in previous research, a data matrix was constructed using observations from different channels over a selected domain. The PCA analysis method was used to extract observation information for different spatial scales over this domain, taking advantage of the independence of PCA modes. The observation information of a target position at different spatial scales was gradually recovered by making full use of the channel correlation and the spatial continuity of observations. The newly proposed PCA iteration method could recover observations

affected by RFI with high precision. The results of the ideal experiment and the real data restoration experiment proved that the accuracy and effectiveness of the new method are much better than those of the Cressman method. Furthermore, the spatial continuity of observations in the recovered data was very well preserved by the new method.

One-week validation results also proved that the new method had good stability and prospects for long-term RFI data restoration, which will be the next step of our research. In addition, future research should consider how to use the recovered data in data assimilation and retrieval research as well as the impact of the accuracy of the recovered data.

**Author Contributions:** Conceptualization, Z.Q. and W.S.; Methodology, Z.Q.; Software, W.S.; Validation, W.S., Z.Q. and Z.L.; Formal analysis, Z.Q.; Investigation, W.S.; Resources, Z.Q.; Data curation, Z.L.; Writing—original draft preparation, W.S.; Writing—review and editing, Z.Q.; Visualization, W.S.; Supervision, Z.L.; Project administration, Z.L.; Funding acquisition, Z.Q.

**Funding:** This research was funded by the National Key R&D Program of China, grant number 2016YFC0402702 and by the Mathematical Theories and Methods of Data Assimilation supported by National Natural Science Foundation of China, grant number 91730304.

**Acknowledgments:** We would like to thank the two anonymous reviewers and editor Bob Du for their encouragement and constructive suggestions which helped to improve the quality of the paper significantly. This research was funded by the National Key R&D Program of China, grant number 2016YFC0402702, 2018YFC1507302, and the Mathematical Theories and Methods of Data Assimilation supported by National Natural Science Foundation of China (Grant No. 91730304).

**Conflicts of Interest:** We declared that we have no conflicts of interest to this work. We declare that we do not have any commercial or associative interest that represents a conflict of interest in connection with the work submitted.

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
