# Peer review of "A New Restoration Method for Radio Frequency Interference Effects on AMSR-2 over North America"

_remotesensing, doi:10.3390/rs11242917_

Round 1
Reviewer 1 Report
The paper "A New Restoration Method for Radio Frequency Interference Effects on AMSR-2 over North America" presents a novel approach to estimating the brightness temperatures of RFI-affected radiometer measurements using the spatial-spectral variation of surrounding non-corrupted measurements. The method appears sound and well presented.
I am including comments for minor updates that will help to strengthen the paper.
The paper inherently assumes that there is very little unique information in a single measurement that is not extractable from the surrounding measurements and the other channels. I would make it clear up front that you are not trying to correct RFI contamination, but are using redundant information to estimate the Tb of the corrupted measurement.
pg 1 line 32: "comes from the ground" -> "comes from the surface".
pg 1 line 36: Remove AMSR-E from the list of "current" radiometers. Add GMI to the list.
pg 4 line 165: Adequately detecting the location of RFI is a big assumption of this paper. I would suggest adding a couple sentences describing what the method would do if RFI is not adequately detected. For example, the RFI from surrounding unflagged cells would corrupt the restoration of the cell of interest.
pg 5 line 185: Please add a reference for PCA.
pg 5 line 185: Add a clarification that the PCA is computed for each contaminated pixel using the surrounding measurements.
Pg 5-6: Although the method presented should be sound, I am concerned that recomputing the PCA multiple times per contaminated pixel would be very computationally expensive. Since you are using several hundred surrounding measurements to form the PCA, I would doubt that including the pixel of interest has much influence on the PCA at all. I would think that you should be able to just do the PCA once, and then use that PCA for every iteration. You don’t have to modify the paper, but it is something you should investigate.
In addition, the iterative method that you are using attempts to give most weight to the highest principle components, but is somewhat ad-hoc. You may be able to get rid of the iterations all-together by using a weighted least-squares fit to the PCA modes. The weights would be determined by the singular values.
Pg 6 Line 222: Here you show the RFI contamination being a single measurement; however, typically Rfi affects a neighborhood of surrounding cells. I suggest doing this test where the neighborhood around the RFI contaminated measurement is also affected.
Pg 6 Line 235: It is unclear whether you iterate until all modes are accounted for, or if you stop at the second mode. Please clarify.
Pg 7 Line 261: “The reconstruction of individual pixels may be contingent”, on what? Please fix.
Pg 8 Line 265: In this test using 36 GHz, what did you assume for the regions contaminated by RFI? Or did you assume that no surrounding pixels were contaminated? This is important, because when it is applied back to 6 GHz, many of the surrounding measurements would be contaminated, and therefore not available for use in the PCA.
Pg 10 Figure 9: This figure is somewhat redundant with the previous two. Suggest removing.
Reviewer 2 Report
The PCA Iterative Method seems like a very good RFI "cleaning" approach for multi-band passive Microwave Radiometers and much better than the Cressman interpolation method it is compared to (as shown by the scatter and PDF plots).
I miss certain points in the analysis though:
It would be interesting to compare the algorithm to other RFI removal techniques and demonstrate whether it is better in all cases or what are the advantages/disadvantages in each case The impact of surrounding pixels selection strategy and the order in which pixels are "cleaned" from the image is also not clearly described, and this could affect the final approach methodology Finally, it would have been interesting to see the final results on all channels and polarisations of the images analysis (and compared the cleaned images overall)Some minor comments as well:
Typo on line 29 "development t of" Figure 4 is not clearly understood, images c and d appear to be the same
Author Response
Please see the attachment.

This manuscript is a resubmission of an earlier submission. The following is a list of the peer review reports and author responses from that submission.